# A Multi-Dimensional Goal Aircraft Guidance Approach Based on Reinforcement Learning with a Reward Shaping Algorithm

**DOI:** 10.3390/s21165643

**Published:** 2021-08-21

**Authors:** Wenqiang Zu, Hongyu Yang, Renyu Liu, Yulong Ji

**Affiliations:** 1College of Computer Science, Sichuan University, Chengdu 610065, China; 2019223045134@stu.scu.edu.cn (W.Z.); yanghongyu@scu.edu.cn (H.Y.); 2015141082046@stu.scu.edu.cn (R.L.); 2School of Aeronautics and Astronautics, Sichuan University, Chengdu 610065, China

**Keywords:** reinforcement learning, aircraft guidance, reward shaping, 4D waypoint navigation

## Abstract

Guiding an aircraft to 4D waypoints at a certain heading is a multi-dimensional goal aircraft guidance problem. In order to improve the performance and solve this problem, this paper proposes a multi-layer RL approach. The approach enables the autopilot in an ATC simulator to guide an aircraft to 4D waypoints at certain latitude, longitude, altitude, heading, and arrival time, respectively. To be specific, a multi-layer RL approach is proposed to simplify the neural network structure and reduce the state dimensions. A shaped reward function that involves the potential function and Dubins path method is applied. Experimental and simulation results show that the proposed approach can significantly improve the convergence efficiency and trajectory performance. Furthermore, the results indicate possible application prospects in team aircraft guidance tasks, since the aircraft can directly approach a goal without waiting in a specific pattern, thereby overcoming the problem of current ATC simulators.

## 1. Introduction

Aircraft guidance [1,2,3,4], especially high-dimensional aircraft guidance, has gradually emerged as a significant research focus in academic circles, owing to the application prospects in complex flight tasks and under realistic conditions. In military exercises, pilots usually need to fly to a series of 4D waypoints [5], which are arranged by air traffic controllers (ATCOs) in advance when performing complex flight tasks. For example, in team aircraft landing tasks, aircraft will sequentially arrive at landing 4D waypoints, and the arrival heading angle of the aircraft is required. The aircraft guidance becomes complicated when the arrival heading angle is taken into account, especially when the arrival time is also considered. Thus, it is essential to seek an approach to solve the multi-dimensional goal aircraft guidance problem, so as to guide an aircraft to 4D waypoints at a certain heading.

Researchers have made significant contributions to aircraft guidance. New local quadratic-biquadratic quality functions [6] were used to obtain more general linear-cubic control laws for aircraft guidance. For altitude and position control, in a previous study [7], incremental nonlinear dynamic inversion control was proposed, which is able to track the desired acceleration of the vehicle across the flight envelope. In another previous study [8], a visual/inertial integrated carrier landing guidance algorithm was proposed for aircraft carrier landing, of which the simulation results showed satisfactory accuracy and high efficiency in carrier landing guidance.

The aforementioned prior research is of positive significance in aircraft guidance. In the actual flight, pilots conduct flight tasks with rich experience and skills. However, in ATC simulators, the aircraft is controlled by an autopilot, not by a real pilot, who is usually not well trained to guide the aircraft on 4D flight tasks. The main issue is that autopilot is unable to generate a 4D trajectory [9,10] to meet the requirement of the multi-dimensional goal task, wherein 4D waypoints of certain latitude, longitude, altitude, and heading and arrival time must be reached.

Consequently, ATCOs call for an intelligent approach to solve the aforementioned problem. In this paper, a possible reinforcement learning (RL) [11] approach with a shaped reward function is proposed to achieve the multi-dimensional goal aircraft guidance task, by formulating the problem as a Markov decision process problem.

RL solves sequential decision-making problems by iteratively estimating value functions and optimal control strategies, which represents the long-term optimal performance of the system. Much attention has been shifted towards RL owing to the performance thereof in a wide range of applications. Thus, the capabilities of RL have stimulated research on aircraft guidance tasks.

As a result of RL development, researchers have proposed deep reinforcement learning (DRL) approaches to solve the problems of aircraft guidance. In a previous study [12], to solve the aircraft sequencing and separation problem, the author explored the possibilities of applying RL techniques for ’time in trail’ tasks. A similar approach was proposed in another study [13], which used DRL to train the aircraft by heading commands and constant speed to guide the aircraft. A trajectory generating method was proposed by using a DQN algorithm to perform a perched landing on the ground [14]. In the above DQN algorithm, noise is considered by the model, which is more in line with the actual scenario in the training process.

In previous studies, RL has been shown to have beneficial application prospects in aircraft guidance. However, some results are inconsistent and further research is required for verification, and, at the same time, there are still a number of limitations. First, RL is a method of constantly trying and exploring from the environment, wherein the complexity of state space will directly affect the difficulty of the task. In prior research, aircraft heading was not taken into account in guidance, which reduced the convergence difficulty due to low-level dimensional state space. When aircraft heading and velocity are considered, convergence is usually difficult as aircraft guidance tasks are performed in a high-dimensional state space. Second, the reward function [15,16,17] is vital and will directly affect the converge efficiency; however, the reward function is usually hard to define and needs reward shaping methods.

The high-dimensional state space has an effect on RL training efficiency, which will lead to a considerable amount of calculations and sparse rewards. To reduce the state space dimensions and difficulties, the multi-level hierarchical RL method and nested policies [18] were proposed. Researchers [19] proposed a nested RL model capable of determining both aircraft route and velocity, using an air traffic controller simulator created by NASA. The latter was employed as a testing environment to evaluate RL techniques, to provide tactical decision support to an air traffic controller, to select the proper route, and to change the velocity for each aircraft. Ultimately, RL methods were evaluated in the aforementioned testing environment to solve the autonomous air traffic control problem for aircraft sequencing and separation. The results revealed that, in the whole training process, the total score tended to oscillate and rise, which limited the application of the above method in practice. Another disadvantage of this approach is that it restricts the position of the aircraft in a fixed place and moves the aircraft in a limited route without considering the effect of aircraft aerodynamics on the flight path.

To design an effective reward function through reward shaping algorithms, it is necessary to speed up the convergence. Two types of reward functions [20] are proposed to assist ATCOs in ensuring the safety and fairness of airlines, by solving the problems of both holding on ground and in air. To solve the problem of aircraft guidance, a new reward function was proposed in [21], to improve the performance of the generated trajectories and the training efficiency.

Recent RL development for nonlinear control systems has implications for aircraft guidance tasks. A Virtual State-feedback Reference Feedback Tuning (VSFRT) method [22] was applied to unknown observable systems control. In [23], a hierarchical soft actor–critic algorithm was proposed for task allocation which significantly improved the efficiency of the intelligent system. In another study [24], a strategy based on heuristic dynamic programming (HDP) (λ) was used to solve the event-triggered control problem in a nonlinear system and improve the system stability, where the one-step-return value was approximated by an actor–critic neural network structure.

In the present paper, a multi-layer RL approach with a reward shaping algorithm is proposed for the multi-dimensional goal aircraft guidance flight task, wherein an aircraft is guided to waypoints at certain latitude, longitude, altitude, heading angle, and arrival time. In the proposed approach, a trained agent is adopted to control the aircraft by selecting the heading, changing the vertical velocity, and altering the horizontal velocity, based on an improved multi-layer RL algorithm with a shaped reward function. The present solution can solve the aircraft guidance problem intelligently and efficiently, and thus is applicable in a continuous environment where an aircraft moves in a continuous expanse of space.

The key contributions of the proposed deep RL approach are multifold:

a. A multi-layer RL model and an intelligent aircraft guidance approach are presented to perform the multi-dimensional goal aircraft guidance flight task, by reducing the state space dimensions and simplifying the neural network structure.

b. A shaped reward function is proposed to enhance the performance of aircraft trajectory, while considering Dubin’s path method.

c. The proposed work provides possible application prospects for the research on aircraft guidance while considering arrival time.

The remainder of the present study is organized as follows: in Section 2, the background concepts on Dubins path and RL are introduced, along with the variants used in the present work; in Section 3, the RL formulation of the aircraft guidance task is presented; in Section 4, the environment settings and structure of model are introduced in detail; in Section 5, numerical simulation results and discussion are given; and, in Section 6, the conclusions of the present study are provided.

## 2. Background

### 2.1. Dubins Path

The Dubins path [25,26,27] is the shortest path between any two configurations, and can be more precisely characterized as: RSR,LSL,RSL,LSR,RLR, and LRL, where *L* denotes “turn left”, *R* denotes “turn right”, and *S* denotes “go straight”. The six classes of Dubins path mentioned can be divided into CSC and CCC curves, which are shown in Figure 1.

For *CSC* curves, the length of the shortest path is defined as:(1)$$L_{csc}=C_{\alpha }+S_{d}+C_{\gamma }$$

For *CCC* curves, the length of the shortest path is defined as:(2)$$L_{CCC}=C_{\alpha }+C_{\beta }+C_{\gamma }$$

### 2.2. Reinforcement Learning

#### 2.2.1. Basics of Reinforcement Learning

RL research belongs to the category of Markov decision process (MDP) [28], which attempts to solve the problem of decision optimization and can be defined as M=(S,A,P,γ,R), where *S* is the set of environment states; *A* is all possible actions the agent can select from the environment; *R* is the set of obtained rewards from the environment; *P* is the transition probabilities function; γ is the discount factor that determines the contribution of future rewards. $s_{t},a_{t},p_{t}$, and $r_{t}$ respectively represent the current state, selected action, transition probability, and reward obtained from the environment.

RL is a method to maximize long-term rewards during interaction with the environment. At one time step, the agent selects an action $a_{t}$ according to the state $s_{t}$. Subsequently, the agent gains a reward $r_{t}$ and steps to the next state $s_{t+1}$. The state-value function V(s) is used to estimate the long-term rewards. The updating of V(s) can be defined as:(3)$$V\left(s_{t}\right)\leftarrow V\left(s_{t}\right)+\alpha \left(r_{t}+\gamma V\left(s_{t+1}\right)-V\left(s_{t}\right)\right)$$
where α is the learning rate.

#### 2.2.2. Policy-Based RL

In the value-based RL approach [29], the agent selects an action to maximize the value function using greedy strategy. The strategy is a mapping from state space to action space, which is the optimal strategy.

Policy-based RL [30,31] is to parameterize the policy $\pi_{\theta }\left(a_{t}|s_{t}\right)$, where θ is the parameter of policy neural network. Then, the total rewards can be defined as:(4)$$G=E\left[\sum_{t=0}^{T}r_{t}|\pi_{\theta }\left(a_{t}|s_{t}\right)\right]$$

Policy-based RL adopts parameterized linear function and nonlinear function (such as neural network) as the strategy. The optimal policy can be defined as:(5)$$\pi^{*}=\underset{\pi }{\arg\max}\left[\sum_{t=0}^{T}r_{t}|\pi_{\theta }\left(a_{t}|s_{t}\right)\right]$$
where *T* is the total time.

In the policy-based RL method, a critic network $V_{\omega }\left(s\right)$ and a policy network $\pi_{\theta }\left(a|s\right)$ are adopted, where ω and θ are the parameters. Critic network is to evaluate the current policy $\pi_{\theta }\left(a|s\right)$. The strategy is directly iterated by iteratively updating critic parameters ω and policy parameters θ. The critic network updates ω through minimizing $L_{\omega }$, which is the expected square error of new estimate value function $r_{t}+\gamma V_{\omega }\left(s_{t+1}\right)$ and old estimate value function $V_{\omega }\left(s_{t}\right)$, which is:(6)$$L_{\omega }=E\left[{\left(r_{t}+\gamma V_{\omega }\left(s_{t+1}\right)-V_{\omega }\left(s_{t}\right)\right)}^{2}\right]$$

The policy network maximizes $J_{\theta }$ which includes advantage function $r_{t}+\gamma V_{\omega }\left(s_{t+1}\right)-V_{\omega }\left(s_{t}\right)$ and entropy regularization term $H\left(\pi_{\theta }\left(a_{t}|s_{t}\right)\right)$ to obtain maximum long-term rewards. $J_{\theta }$ can be defined as:(7)$$J_{\theta }=E\left[\log\pi_{\theta }\left(a_{t}|s_{t}\right)\left(r_{t}+\gamma V_{\omega }\left(s_{t+1}\right)-V_{\omega }\left(s_{t}\right)\right)+\beta H\left(\pi_{\theta }\left(a_{t}|s_{t}\right)\right)\right]$$
where ω is the parameter of value function $V,H\left(\pi_{\theta }\left(a_{t}|s_{t}\right)\right)$ is the entropy regularization term which represents $-\pi_{\theta }\left(a_{t}|s_{t}\right)\log\pi_{\theta }\left(a_{t}|s_{t}\right)$, the strategy used for encouraging exploration and preventing premature convergence to sub-optimal polices, and β is the coefficient.

## 3. RL Formulation

To guide aircraft to the 4D waypoints at a certain heading, it is necessary to design 4D waypoints series and guide the aircraft to 4D waypoints at a certain heading.

### 3.1. 4D Waypoints Design

4D waypoints are waypoints with attributes of coordinates (latitude, longitude, altitude) and arrival time, as shown in Figure 2, wherein an aircraft is flying through a series of 4D waypoints. A 4D waypoint can be defined as: $P_{k}=\left(x_{k},y_{k},z_{k},t_{k}\right)$, where $x_{k},y_{k},z_{k},t_{k}$ are the longitude, latitude, altitude, and arrival time of aircraft at 4D waypoint $P_{k}$, respectively.

Assuming that there are n+1 4D waypoints on the flight route from the departure airport $P_{n}$ to the arrival airport $P_{0}$, which is shown in Figure 3, the 4D waypoints are sequenced and numbered from the arrival airport to the departure airport: $P_{L}=\left\{P_{0},P_{1},P_{2},P_{3},\ldots P_{n}\right\}$. The n+1 4D waypoints divide the route *L* into *n* segments, $L=\left\{L_{i},\;i=0,1,2,\ldots ,n-1\right\}$, where $L_{i}$ is the segment between 4D waypoint $P_{i}$ and 4D waypoint $P_{i+1}$. The time can be calculated from each position to the landing site $P_{0}$, respectively: $E_{i}=\left\{E_{1},E_{2},E_{3},\ldots ,E_{n}\right\}$.

The time from 4D waypoint $P_{i}$ to 4D waypoint $P_{0}$ is:(8)$$E_{i}=\int_{P_{i}}^{P_{0}}\frac{1}{v_{i}}d\vec{s}$$
where $\vec{s}$ is the distance and $v_{i}$ is the velocity of aircraft.

The flight segment $L_{i}$ can only be a straight line or an arc. The attribute of $L_{i}$ is determined by the two 4D waypoints $P_{i}$ and $P_{i+1}$, and the connection attribute $R_{i}$ between them, where $R_{i}$ is the radius of the flight segment between $P_{i}$ with $P_{i+1}$. Let $S_{i}$ represent the distance from $P_{i}$ to $P_{i+1}$, and $L_{i}$ can be defined as $L_{i}=\left\{P_{i},P_{i+1},S_{i},R_{i}\right\}$. When $R_{i}=0$, the flight segment is a straight line, whereas, when $R_{i}>0$, the flight segment is an arc. Each 4D waypoint of the route has unique inherent attributes: $P_{i}=\left(x_{i},y_{i},z_{i},t_{i}\right)$, which respectively represent the longitude, latitude, altitude, and arrival time. The arrival time $t_{i}$ is the time when aircraft arrives at $\left(x_{0},y_{0},z_{0}\right)$ from current position $\left(x_{i},y_{i},z_{i}\right)$.

When the flight segment $L_{i}$ is a straight line, the radius $R_{i}$ is 0. The length of the flight segment $L_{i}$ is:(9)$$S_{i}=\sqrt{{\left(X_{i}-X_{i+1}\right)}^{2}+{\left(Y_{i}-Y_{i+1}\right)}^{2}+{\left(Z_{i}-Z_{i+1}\right)}^{2}}$$
where (*X*, *Y*, *Z*) are geocentric coordinates of (*x*,*y*,*z*).

When the flight segment $L_{i}$ is an arc:  $L_{i}=\left\{P_{i-1},P_{i},S_{i},R_{i}\right\}$, where $R_{i}>0$. $L_{i}$ will be determined by four known points: $P_{i-1}\left(x_{i-1},y_{i-1},z_{i-1},t_{i-1}\right),P_{i}\left(x_{i},y_{i},z_{i},t_{i}\right),P_{i+1}\left(x_{i+1},y_{i+1},z_{i+1},t_{i+1}\right)\;$ and $P_{i+2}(x_{i+2},y_{i+2}$, $z_{i+2},t_{i+2})$. Segments $L_{i-1}$ and $L_{i+1}$ are straight line segments, and $L_{i}$ is a circular arc with point *O* as the center and $R_{i}$ as the radius. θ is the central angle of the arc.

The length of the flight segment $L_{i}$ is:(10)$$S_{i}=\sqrt{{\left(\frac{\theta \pi R_{i}}{180^{\circ }}\right)}^{2}+{\left(Z_{i}-Z_{i+1}\right)}^{2}}$$

From (8)–(10), an observation can be made that, when the velocity distribution of a route $L=\left\{P_{i},i=0,1,2\ldots ,n\right\}$ is determined, the arrival time of an aircraft moving from 4D waypoint $P_{i}$ to 4D waypoint $P_{0}$ can be determined as:(11)$$t_{i}=\left\{t_{0}+E_{i},i=1,2,3,\ldots ,n\right\}$$
where $t_{0}$ is the current time of $P_{0}$. The interval Δt between 4D waypoints is determined according to the actual airline requirements. Figure 4 shows a series of 4D waypoints. For example, if the current time at $P_{0}$ is 08:00:00 and Δt is 2 min, the arrival time at $P_{0}$ of an aircraft departure from $P_{6}$ can be calculated by (11), which is 08:12:00.

### 3.2. Fly to Waypoints

Flying to 4D waypoints is an aircraft guidance problem. Figure 5 shows the kinematic model of the aircraft, wherein an agent guides an aircraft from a current position $\left(x_{a},y_{a},z_{a},\chi_{a},t_{a}\right)$ to the target position $\left(x_{g},y_{g},z_{g},\chi_{g},t_{g}\right)$. x,y,z, χ and *t* are the longitude, latitude, altitude, heading angle, and arrival time, respectively. $\varphi_{a}$ is the pitch angle of aircraft. The subscripts *a* and *g* denote the aircraft and goal. During movement, velocity and heading directly affect the position, and thus the kinematic equations and mathematical relationship can be defined as:(12)$$\left\{\begin{array}{c} \dot{x}=\frac{v\cos\varphi \sin\chi }{\left(R_{e}+z\right)\cos y}\\ \dot{y}=\frac{v\cos\varphi \cos\chi }{\left(R_{e}+z\right)}\\ \dot{z}=v\sin\varphi \end{array}\right.$$
where $\dot{x},\dot{y}$, and $\dot{z}$ are the delta of the 3D coordinates of the aircraft, *v* is the velocity, φ is the pitch angle, χ is the heading angle (with respect to the geographical north) of the aircraft, $R_{e}$ is the Earth radius, and *z* is the current altitude of aircraft (with respect to sea level).

In the actual environment, the above variables can be obtained by multi sensors; however, in ATC simulators, the aircraft information can be obtained directly without error. (13) defines the change rates of velocity and heading angle:(13)$$\left\{\begin{array}{c} a_{h-\min}<{\dot{v}}_{h}<a_{h-\max}\\ a_{z-\min}<{\dot{v}}_{z}<a_{z-\max}\\ a_{\chi -\min}<\dot{\chi }<a_{\chi -\max} \end{array}\right.$$
where ${\dot{v}}_{h},{\dot{v}}_{z}$, and $\dot{\chi }$ are the delta of horizontal velocity, vertical velocity, and heading angle turn rate, respectively. $a_{h-\min}$, $a_{z-\min}$, $a_{\chi -\min}$, $a_{h-\max}$, $a_{z-\max}$ and $a_{\chi -\max}$ are the minimum and maximum of acceleration of horizontal velocity, vertical velocity, and heading angle, respectively.

For aircraft guidance, the generated trajectory is required to be smooth, and other factors should be considered, which will be analyzed in the reward shaping subsection.

In the present study, the decision-making problem can be formulated as finite horizon MDP, which can be defined as M=(S,A,P,γ,R). *S* and *A* denote the state space and action space, respectively, and both are defined as high-dimensional continuous spaces. For example, the aircraft state $\left(x_{a},y_{a},z_{a},\chi_{a},t_{a}\right)$ and the destination state $\left(x_{g},y_{g},z_{g},\chi_{g},t_{g}\right)$ are composed of 10 dimensions. Generally, the arrival time $t_{g}$ of goal state is zero, whose state space dimension can be reduced to 9. Additionally, by expressing the actions performed by the agent performing the task in terms of heading, velocity, and altitude commands, the action space also becomes a multi-dimensional continuous space. *P* denotes the state transition probabilities function. γ is the discount factor. *R* denotes the set of rewards that the agent obtains from the environment.

Figure 6 shows the flow chart of the training process. The total steps from when the aircraft starts from the initialization state to the termination state are referred to as an episode. In the initialization process, information such as the position and movement model of the aircraft and the goal state are initialized, in addition to the reward shaping value, which will be explained in detail in the next section. After the initialization of the environment, in each step that does not reach the termination state, the agent selects an action $a_{t}$ from current state $s_{t}$, then the environment steps in next state $s_{t+1}$ and returns the reward $r_{t}$. The tuple $\left(s_{t},a_{t},r_{t},s_{t+1}\right)$ is stored into the replay buffer until the environment reaches the termination state to update the policy while training the agent.

### 3.3. Training Optimization

#### 3.3.1. Multi-Layer RL Algorithm

In consideration of the flight task as a multi-dimensional goal task, it is necessary to involve the multi-layer RL algorithm. The multi-layer RL algorithm can divide the flight task into several sub tasks, which also decreases both the dimensions of state space and action space. The sub control layers are divided as follows:Position control layer: control the heading angle of aircraft;Altitude control layer: control the vertical velocity of aircraft;Velocity control layer: control the horizontal velocity of aircraft.

The three sub control layers have their own structure, which can be seen as three single neural networks and are integrated into a main neural network. Sub layers are updated by updating the main neural network, so they run sequentially. Figure 7 shows the framework of the algorithm, from which it can be seen that the main neural network consists of “Actor Model” and “Critic Model”. Both of the two models have three sub layers, and they update the parameters in the same training step. The algorithm is shown in Algorithm 1. In every training step of one episode, the agent selects three actions to control position, altitude, and velocity of the aircraft by the “Actor Model”. Then, the environment obtains the selected actions and steps into next state, during which the rewards of three sub control tasks are respectively calculated by (22)–(27). Finally, the agent will learn from the data in replay buffer.
**Algorithm 1** Multi-layer RL algorithm.1:// Assume policy parameters of three sub layers are $\theta_{p}$, $\theta_{z}$ and $\theta_{v}$2:// Assume critic parameters of three sub layers are $\omega_{p}$, $\omega_{z}$ and $\omega_{v}$3:// Assume $r_{t}^{p}$,$r_{t}^{z}$ and $r_{t}^{v}$ are the step rewards of three sub layers respectively4:Initialize the environment5:Initialize parameters $\theta_{p}$, $\theta_{z}$, $\theta_{v}$, $\omega_{p}$, $\omega_{z}$ and $\omega_{v}$6:**if** is training mode **then**7: **for** each episode **do**8:  Randomly initialize the environment parameters9:  **for** each episode in range episodes **do**10:   obtain actions $a_{t}^{p}$ according to $\pi_{\theta_{p}}\left(a_{t}|s_{t}\right)$; let $p_{t}^{p}=\pi_{\theta_{p}}\left(a_{t}|s_{t}\right)$11:   obtain actions $a_{t}^{z}$ according to $\pi_{\theta_{z}}\left(a_{t}|s_{t}\right)$; let $p_{t}^{z}=\pi_{\theta_{z}}\left(a_{t}|s_{t}\right)$12:   obtain actions $a_{t}^{v}$ according to $\pi_{\theta_{v}}\left(a_{t}|s_{t}\right)$; let $p_{t}^{v}=\pi_{\theta_{v}}\left(a_{t}|s_{t}\right)$13:   let $A_{t}=\left(a_{t}^{p},a_{t}^{z},a_{t}^{v}\right)$, $P_{t}=\left(p_{t}^{p},p_{t}^{z},p_{t}^{v}\right)$, $R_{t}=\left(r_{t}^{p},r_{t}^{z},r_{t}^{v}\right)$14:   step the environment and get tuple ($s_{t},A_{t},P_{t},R_{t},s_{t+1}$)15:   store tuple data in replay buffer16:   **if** done **then**17:    break18:   **end if**19:  **end for**20:  update ω in $\left\{\omega_{p},\omega_{z},\omega_{v}\right\}$ by minimizing L(ω) by (7)21:  update θ in $\left\{\theta_{p},\theta_{z},\theta_{v}\right\}$ by maximizing J(θ) in (6)22: **end for**23:**else**24: **for** each testing episode **do**25:  **for** each step in range episodes **do**26:   run the environment27:  **end for**28: **end for**29:**end if**

#### 3.3.2. State Space

In the present experiment, all possible states have an impact on the final results in the RL environment. Therefore, it is important to consider all parameters that may have an impact on the experimental results when setting the state space. The present experimental goal was to reach a target position (latitude, longitude, altitude, and heading) at the correct time. For the above purpose, a multi-layer RL model was introduced with three layers: a position control layer to select the heading, a velocity control layer to change the velocity, and an altitude control layer to alter the aircraft altitude. The state space was designed separately for each of the layers.

For the position control layer, which aimed to lead the aircraft towards the target position (latitude, longitude) having a goal heading, the state space $S_{p}$ was designed as:(14)$$S_{p}=\left\{\Delta x,\Delta y,\chi_{a},\chi_{g}\right\}$$
where Δx denotes the delta longitude of the target and the aircraft Δy denotes the delta latitude of the target and the aircraft, $\chi_{a}$ represents the aircraft heading, and $\chi_{g}$ represents the goal heading. The domain of x,y, and χ are [−180,180],[−90,90] and [−180,180], in degrees, respectively.

For the velocity control layer, which aimed to reach the target position (latitude, longitude, and heading) at certain time, the state space $S_{v}$ was designed as:(15)$$S_{v}=\left\{\Delta d\right\}$$
(16)$$\Delta d=v_{a}*t_{a}-d_{a}$$
where $v_{a}$ denotes horizontal velocity of the aircraft, $t_{a}$ is the arrival time, and $d_{a}$ is the distance of aircraft to goal. The domain of $v_{a}$ will be introduced in the “Numerical Experiment” section.

For the altitude control layer, which aimed to reach the target altitude, the state space $S_{z}$ was designed as:(17)$$S_{z}=\left\{\Delta z\right\}$$
(18)$$\Delta z=z_{a}-z_{g}$$
where Δz is the delta of $z_{a}$ and $z_{g}$, $z_{a}$ denotes the current altitude, and $z_{g}$ denotes the goal altitude. The domain of the altitude *z* is from 0 m to 10,000 m.

#### 3.3.3. Action Space

In the multi-layer RL algorithm, three layers that output actions, heading angle, vertical velocity, and horizontal velocity, respectively, were included.

The action space of the position control layer can be defined as:(19)A={0,1,2}
where 1 means the aircraft remains at the current heading; 0 and 2 represent the left turn and right turn of the aircraft, respectively.

The action space of the vertical velocity control layer can be defined as:(20)A={0,1,2}
where 1 means the aircraft remains at the current altitude and the vertical velocity is zero; 0 and 2 represent descending and climbing, respectively.

The action space of the horizontal velocity control layer can be defined as:(21)A={0,1,2}
where 1 means the aircraft remains at the current horizontal velocity; 0 and 2 represent deceleration and acceleration, respectively.

#### 3.3.4. Termination State

The environment resets when entering a termination state. The following termination states were designed:Running out of time. The agent is trained every 300 steps, and, if the agent is trained for more than 300 steps, time runs out and the environment resets.Reaching the goal. If the agent-goal distance is less than 2 km and the delta of the heading angle is lower than 28°, the aircraft is assumed to have reached the goal.

#### 3.3.5. Reward Function Design

To learn a policy for an MDP M=(S,A,P,γ,R), the reinforcement learning algorithm could instead be run on a transformed MDP.

$M^{\prime }=\left(S,A,P,\gamma ,R^{\prime }\right)$, where $R^{\prime }=R^{\prime }+F$ is the transformed reward function, and F:S×A×S→R is the shaping reward function.

Potential function ϕ(s) [32,33] is possible applied to reward shaping, which will modify the reward function to accelerate the agent’s learning to move straight forward to the goal. For each state *s*, we added the difference of potentials to the reward of a transition. Figure 8 shows an agent learning to reach goal, with a +3 reward for going up to a higher potential value state, a −3 reward for going down to a lower potential value state, and an additional −1 reward for losing time at each step.

A shaping reward function F:S×A×S→R is potential-based if there exists ϕ:S→R:(22)$$F\left(s_{t},a_{t},s_{t+1}\right)=\gamma \phi \left(s_{t+1}\right)-\phi \left(s_{t}\right)$$

Owing to *F* being a potential-based shaping function, every optimal policy in $M^{\prime }=\left(S,A,P,\gamma ,R^{\prime }\right)$ will also be an optimal policy in M=(S,A,P,γ,R).

At every step, the agent takes an action $a_{t}$ from current state $s_{t}$ and transits to the next state $s_{t+1}$. The reward function can be defined as:(23)$$R\left(s_{t},a_{t},\;s_{t+1}\right)=F\left(s_{t},a_{t},\;s_{t+1}\right)+T\left(s_{t+1}\right)$$
where $T(s_{t+1})$ is the terminal state reward, which is defined as:(24)$$T\left(s_{t+1}\right)=\left\{\begin{array}{c} 500,\;\mathrm{if}\;\mathrm{reach}\;\mathrm{goal}\;\\ -10,\;\mathrm{if}\;\mathrm{run}\;\mathrm{out}\;\mathrm{of}\;\mathrm{time}\;\\ 0,\;\mathrm{else}\; \end{array}\right.$$

The aircraft guidance task involves five demands: latitude, longitude, altitude, arrival time, and heading angle. Thus, $\phi \left(s_{t}\right)$ is defined as:(25)$$\phi \left(s_{t}\right)=D\left(s_{t}\right)+0\left(s_{t}\right)+H\left(s_{t}\right)+A\left(s_{t}\right)$$
(26)$$\left\{\begin{array}{c} D\left(s_{t}\right)=-e_{D}\sqrt{{\left(x_{a}-x_{g}\right)}^{2}+{\left(y_{a}-y_{g}\right)}^{2}}\\ O\left(s_{t}\right)=-e_{o}\left|\chi_{a}-\chi_{g}\right|\\ H\left(s_{t}\right)=-e_{H}\left|z_{a}-z_{g}\right|\\ A\left(s_{t}\right)=-e_{A}\left|t_{a}-L_{d}/v_{a}\right| \end{array}\right.$$
where $D\left(s_{t}\right)$ is the horizontal distance reward function, which denotes the distance to the target. $O\left(s_{t}\right)$ is the direction reward function, which denotes the heading angle to the target; $H\left(s_{t}\right)$ is the altitude reward function, which denotes the distance in altitude to the target; $A\left(s_{t}\right)$ is the arrival time reward function, which denotes the arrival time to the target; $L_{d}$ is the length of Dubins path and can be calculated by (1) and (2); $e_{D},e_{O},e_{H}$ and $e_{A}$ are coefficients.

However, too many factors considered in the reward function will lead to low convergence efficiency and local region of application. There are four sub functions in (25), and in the present study, a shaped reward function was proposed, wherein the reward function form was simplified using the multi-layer reward function.

In (25), $D\left(s_{t}\right)$ and $O\left(s_{t}\right)$ are merged into one function. Thus, $\phi \left(s_{t}\right)$ can be redefined by three parts:(27)$$\left\{\begin{array}{c} P\left(s_{t}\right)=-e_{P}L_{d}\\ H\left(s_{t}\right)=-e_{H}\left|z_{a}-z_{g}\right|\\ A\left(s_{t}\right)=-e_{A}\left|t_{a}-L_{d}/v_{a}\right| \end{array}\right.$$
where $e_{P}$ is the coefficient. Table 1 shows all the coefficients of reward functions.

## 4. Numerical Experiment

In this section, we will first describe the experiment setup. Then, the training model will be introduced in detail.

### 4.1. Experiment Setup

In the present study, three guidance simulation experiments were conducted and compared. Firstly, an experiment was conducted, wherein four models guided aircraft to 3D waypoints (heading angle was also considered) in a constant velocity without considering the arrival time of the aircraft. The four models were: a multi-layer model without reward shaping, a multi-layer model with reward shaping, a not layered model without reward shaping and a not layered model with reward shaping, respectively. For comparison, the not layered model did not have sub control layers and directly selected a three-dimensional vector as heading action, vertical velocity action, and horizontal velocity action, respectively, where the reward function was also not layered. Secondly, arrival time and velocity changes were considered and the models guided aircraft to 4D waypoints (heading angle was also considered). Finally, the well trained model was used to verify the team performance by guiding aircraft to a series of 4D waypoints, as shown in Figure 9. In this experiment, three aircraft were guided to 4D waypoints, which were distributed in advance, in the same or different aerospace.

In real air guidance, information of an aircraft is obtained by sensors. In the present study, an aircraft was trained under an ATC simulator, wherein aircraft information could be obtained directly and without error.

The experiment settings and details were as follows:

The experimental environment utilized in the present paper was mainly based on the Bluesky simulator [34], which is an open-source air traffic control simulator using OpenAP [35] aircraft performance models. The training environment was set in ZUUU airport terminal aerospace, located at latitude 30.5635165° North, and longitude 103.939946° East, with the runway oriented at 22°. The scenario involved an F-16 aircraft under training, the task of which was to fly towards the series of 4D waypoints for landing preparation. The initial position of the aircraft was at the ZUUU south aerospace field and the aircraft was to fly towards the north direction.

As shown in Table 2, *x*, *y*, and *z* are longitude, latitude, and altitude of aircraft. The maximum and minimum horizontal velocity were set as 500 km/h and 270 km/h, respectively. As such, the maximum turn rate of the aircraft was set to ±6° per second and the maximum acceleration was set to ±2 m per second. The simulation had a time interval of one second. To speed-up the simulation and effectively process the large amount of sampling data, the maximum simulation speed was used during training.

At the start of an episode, the initialization of the environment involved multiple parameters. The system generated a series of 4D waypoints at the landing direction of the ZUUU airport, and, at the same time, an arrival time sequence was generated. The arrival time should not be too high or too low, which would make the speed of the aircraft to become unreasonably high or low. The initial position of the aircraft was established in the ZUUU south airspace, with a random heading, at horizontal velocity 500 km/h and altitude 1000 m. During the simulation, the arrival time of aircraft guided to the aimed for navigation 4D waypoint decreased as the simulation time evolved. The aircraft guidance was considered to be successful when the distance to the current navigation 4D waypoint was less than 2 km, and the delta heading angle of aircraft and 4D waypoint was less than 28°, where we could use heading commands to control an aircraft to reach goal, or the arrival time to the current navigation 4D waypoint was less than zero.

### 4.2. Models and Training

In this section, the parameters and models are introduced. The time steps *T* were 300 and the mini batch size *M* was 1000, and discount factor γ was 0.98. Table 3 lists all the parameters.

In the present RL method, a multi-layer RL architecture is exploited to effectively manage the complex air traffic control task examined in the present study. In the next section, the design details and the training process of the multi-layer models will be described.

#### 4.2.1. Models

There are three sub layers (position control layer, velocity control layer, and altitude control layer) which select actions to control the aircraft position, velocity, and altitude.

In the position control layer, the Adam optimizer and mean squared error loss function were adopted to learn the neural network parameters with a learning rate of $5\times 10^{-4}$. The critic network had two hidden layers with 128 and 32 units, adopting ReLU [36] and tanh as activation functions [37], respectively. The policy network had two hidden layers with 64 and 32 units, respectively, with ReLU as the activation function. For the output layer, softmax activation function was adopted. During the training of the position control layer, the rewards obtained from the environment should be preprocessed. At every step, the average reward of all possible actions should be calculated; then, the selected action reward should subtract the average reward and be normalized to the sign value.

Both the velocity control layer and the altitude control layer were simple in policy network structure, with 32 units of the hidden layer. Given the distribution of the output, ReLU and the softmax [38] were chosen as the activation functions, respectively, for the two layers. The critic network contained a hidden layer of 64 units. ReLU activation function was adopted as well. Finally, the three control layers were trained using the gradient descent method.

Appropriately setting the model hyperparameters is a crucial factor for optimal performance. Table 3 lists all the hyperparameters in the present network.

The interplay between the model and the environment were also vital to the experiment. The sampling [39] effect in the training process directly affected the convergence efficiency and thus the performance of the algorithm. The aim of the present study was that every sample included valuable data contributing towards a good solution. Thus, the hindsight experience replay (HER) [40] method was adopted to optimize the randomly sampled data so that the neural network could have a better convergence direction and solve the problem of sparse rewards.

#### 4.2.2. Training

In this paper, the actor–critic RL algorithm is adapted to train the agent. The training process is: at the beginning of every training process, initialize critic network *V* and policy network π with weights ω and θ.

In each iteration, use old policy θ to interact with the environment and calculate loss function by (6), then update the critic network by minimizing loss $L_{\omega }$:(28)$$\omega \leftarrow \omega +\alpha \nabla_{\omega }L$$
where α is the step length of gradient descent.

The policy θ is updated by maximizing $J_{\theta }$ in (7):(29)$$\theta \leftarrow \theta +\beta \nabla_{\theta }J$$
where β is the step length of gradient descent.

## 5. Analysis of Results

In this section, analysis is first conducted on the performance without considering the arrival time and the aircraft being under a constant velocity at 500 km/h. Then, the arrival time is taken into consideration, and the results are analyzed. Finally, the well trained model is used to perform the team aircraft guidance task, and the results are analyzed.

### 5.1. Training Performance

#### 5.1.1. Without Considering Arrival Time

Figure 10 shows the success rates of different approaches during more than 15,000 training episodes. In Figure 10, four success rate curves are presented, which represent the rates of the aircraft reaching the goal, using a multi-layer RL approach without reward shaping, a multi-layer RL approach with reward shaping, a not layered approach without reward shaping, and not layered approach with reward shaping, respectively.

The multi-layer RL approach with reward shaping quickly reached a high success rate at 90% before 8000 training episodes, compared with other approaches. Furthermore, the success rate reached the maximum at around 95% before 14,000 training episodes. Compared with the former approach, without reward shaping, the multi-layer RL approach remained at a low success rate of less than 20%, which suggested a lower efficiency in exploring and exploiting. However, compared with not layered approaches, the multi-layer approach showed a significant performance in the success rate. The curves of the non-layered approaches showed that they cannot achieve the goal without the multi-layer approach because of the sparse rewards and local regions of the applications.

The maximum success rates and average computational times obtained by testing each well trained model in 100 simulations are shown in Table 4. The multi-layer RL approach with reward shaping had the highest maximum success rate of 95%, compared with other approaches. According to the simulation results, the average calculation time of all approaches was less than 10 ms, while the calculation time of the multi-layer approach was longer.

After the four models were trained for 15,000 episodes, we selected the best generated trajectories for comparison when testing each model in 100 simulations. Figure 11 shows the trajectories of the four approaches guiding aircraft to a waypoint at certain latitude, longitude, altitude, and heading, respectively, without considering arrival time. During the guidance, the aircraft maintained a constant horizontal speed.

As shown in Figure 11a, when using a multi-layer approach without reward shaping, the aircraft reached its goal in altitude but kept whirling and failed to reach the goal position. In Figure 11c,d, when the multi-layered approach was not used, the aircraft kept whirling and failed to reach the goal in both altitude and position. Compared with the former approaches, shown in Figure 11b, when using a multi-layer approach with reward shaping, the aircraft successfully reached its goal in both altitude and position, which demonstrated that the performance of aircraft guidance could be significantly improved using a multi-layer approach with reward shaping.

#### 5.1.2. Considering Arrival Time

When considering arrival time, the model action space should take velocity changes into consideration. In this experiment, four approaches were used to guide aircraft to a 4D waypoint at certain latitude, longitude, altitude, heading angle, and arrival time, during which aircraft would alter the horizontal velocity to accommodate the arrival time.

Figure 12 shows the success rates of different approaches during more than 15,000 training episodes. For the approaches considering arrival time, the final success rate decreased as the state space increased. Compared with Figure 10, the curve of the multi-layer approach with reward shaping became unstable. Because of the larger state space, the success rates of other approaches remained low.

As shown in Table 5, compared with approaches without considering arrival time, the final success rates of approaches considering arrival time were 2%, 3%, 1%, and 3% lower, respectively, and the average computational times were slightly increased by 2.1 ms, 2.3 ms, 0.7 ms, and 0.7 ms, respectively.

Figure 13 shows the 4D trajectories of the aircraft. Compared with Figure 11, when considering arrival time, the trajectory was slightly different when the aircraft turned because of the velocity changes and the differences in the turning radius. From Figure 11b and Figure 13b, we can see that the aircraft was farther from the goal when reaching termination state, due to the velocity changes.

### 5.2. Multi Aircraft Performance

In this experiment, three aircraft were involved and well-trained models were used to test the team guidance performance of multi-aircraft. Figure 14 is a color map figure which shows the 2D trajectories and the arrival times of aircraft, where the color map value such as −100 is the delta of current time and final time in seconds. Figure 15 shows the 3D trajectories how the aircraft flew to the aimed for 4D waypoint, where three aircraft started within the predetermined scope in the same aerospace (Figure 15a) and different aerospaces (Figure 15b), and aircraft aimed at the series landing 4D waypoints, respectively. The simulation results are shown in Table 6, both the average distance and delta heading angle were in the tolerances within 2 km and ±28°. The simulation results show that the trajectories can be generated to guide aircraft to a series of 4D waypoints.

Figure 16, Figure 17 and Figure 18 respectively give the changes of heading angle, altitude, and velocity. According to the figures, heading angle and velocity change more frequently and irregularly, which should be avoided in the actual flight. Since the turning radius varies with the velocity, the flight trajectory is a velocity-dependent curve, which is the reason for frequent changes in the velocity. The change of altitude is linear because the goal altitude is definite and invariable.

## 6. Conclusions

In the present study, a deep multi-layer RL algorithm is proposed that can overcome the multi-dimensional goal aircraft guidance problem. In the proposed method, the problem is formulated as an MDP problem, and the aircraft is controlled by selecting the heading, changing the vertical velocity, and altering the horizontal velocity. The results of the present numerical experiments reveal that the proposed algorithm has promising prospects in assisting an aircraft to reach a target 4D waypoint at a certain heading, which can be applied in team aircraft guidance tasks. The advantage of the present method is providing a potential solution enabling autonomous 4D aircraft guidance in a structured airspace.

Additionally, a hierarchical architecture is proposed for a multi-layer RL agent, and the capability thereof was demonstrated to solve multi-goal aircraft guidance decision-making problems. The promising results from the present numerical experiments have provided encouragement to conduct future work on more advanced ATC simulators. However, the present experiment and algorithm have limitations. In the actual flight, the flight trajectory should be smooth and simple, and frequent changes of velocity and heading angle are undesirable. The focus of future work will be on continuous action reward shaping to solve the current constraints.

## Figures and Tables

**Figure 1 sensors-21-05643-f001:**
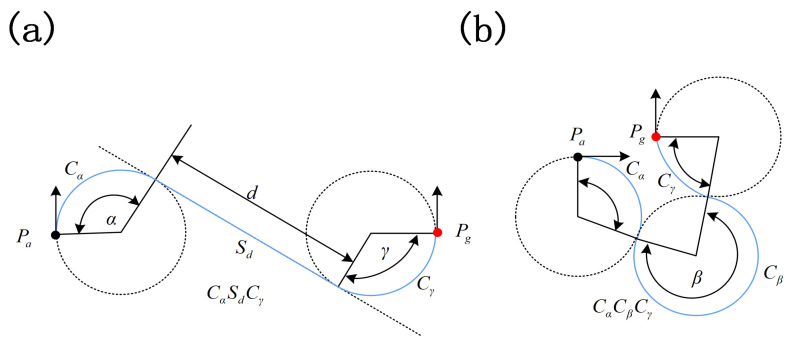
Dubins path: (**a**) *CSC* curve and (**b**) *CCC* curve.

**Figure 2 sensors-21-05643-f002:**
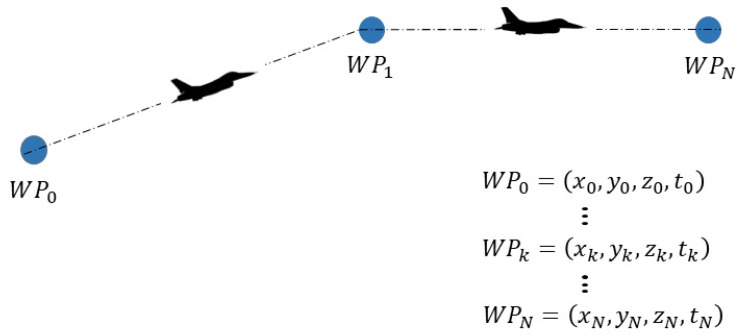
Aircraft fly towards 4D waypoints series.

**Figure 3 sensors-21-05643-f003:**
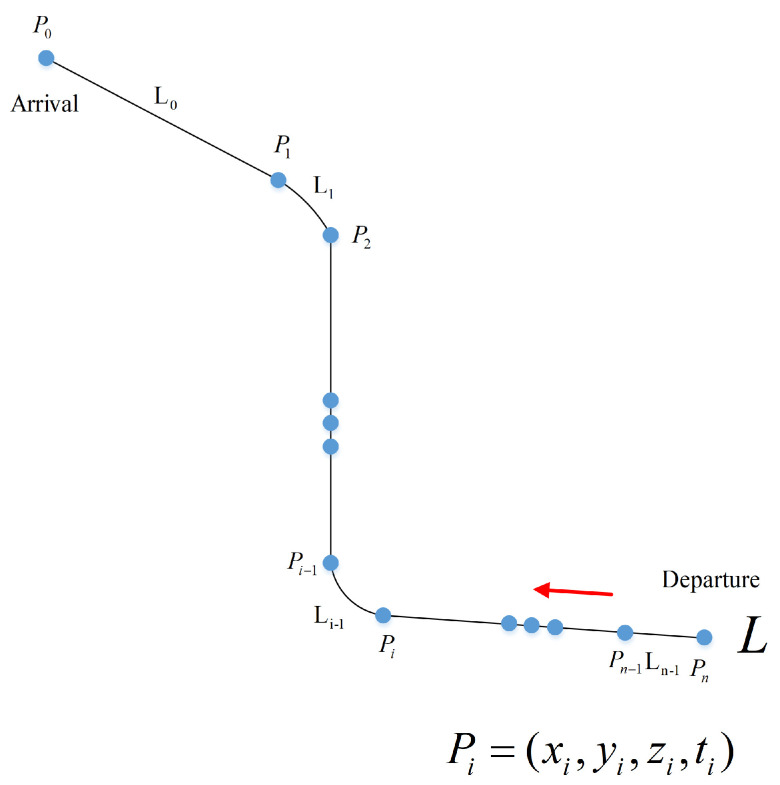
Line and arc flight segments.

**Figure 4 sensors-21-05643-f004:**
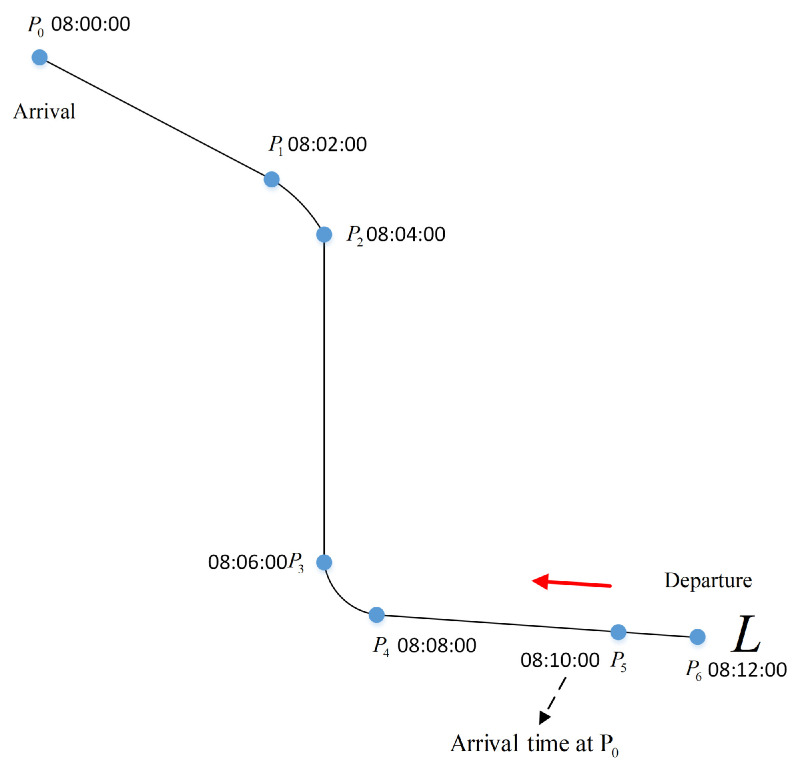
Arrival time of the 4D waypoint.

**Figure 5 sensors-21-05643-f005:**
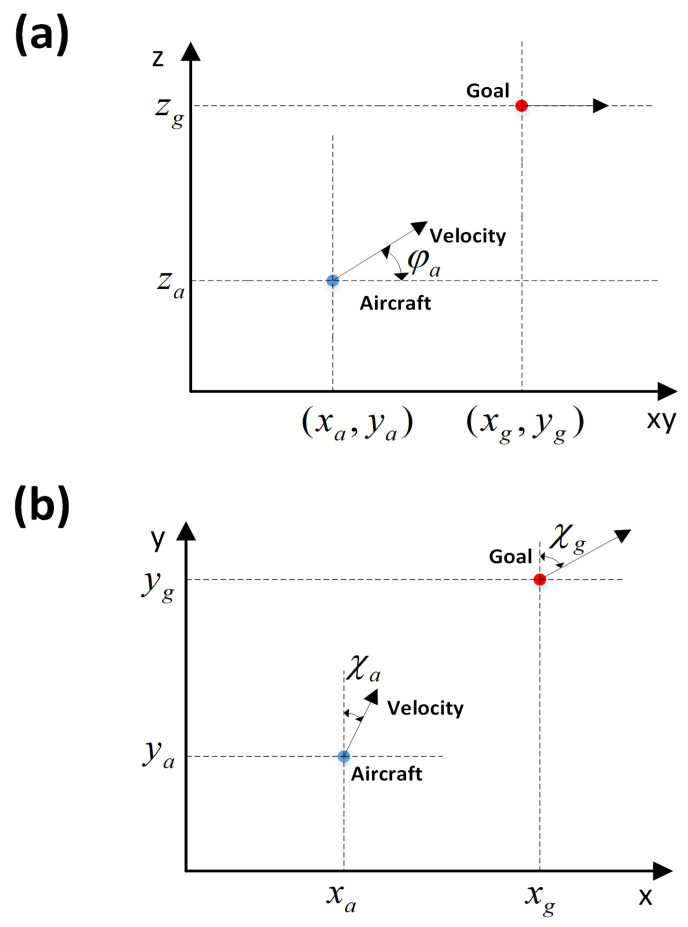
Kinematic model of aircraft: (**a**) kinematic model on a vertical plane and (**b**) kinematic model on a horizontal plane.

**Figure 6 sensors-21-05643-f006:**
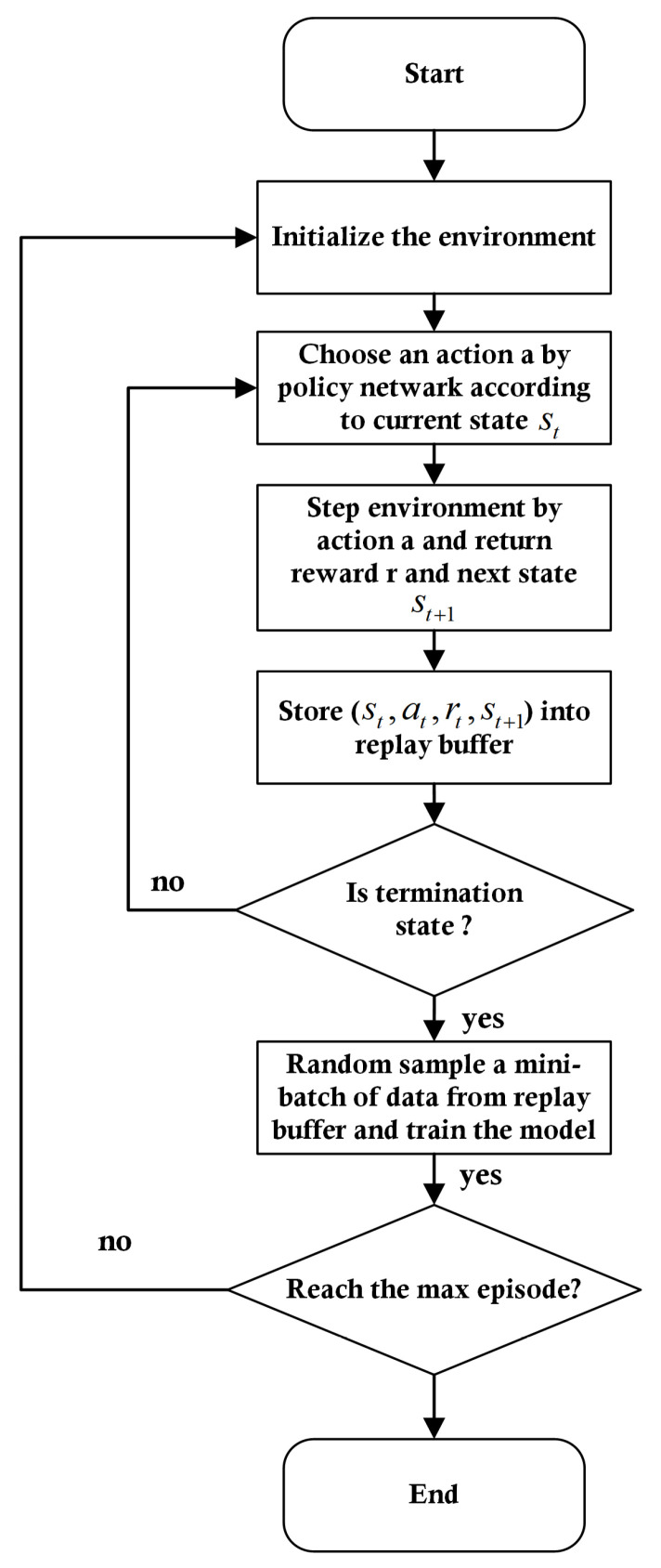
Flow chart of the planning algorithm.

**Figure 7 sensors-21-05643-f007:**
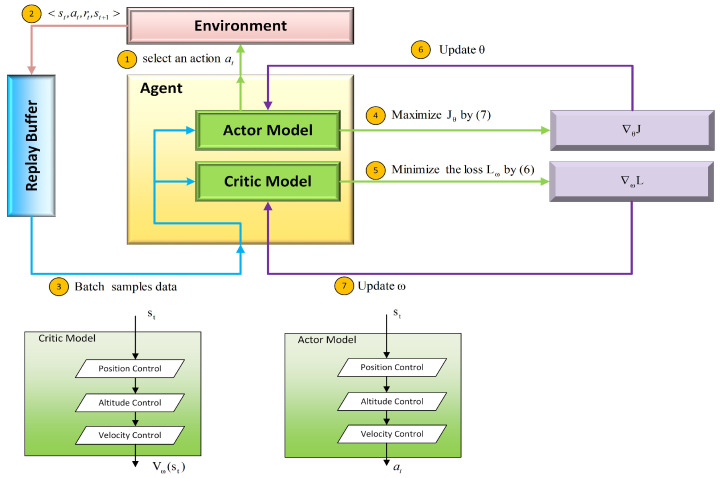
Framework of the planning algorithm.

**Figure 8 sensors-21-05643-f008:**
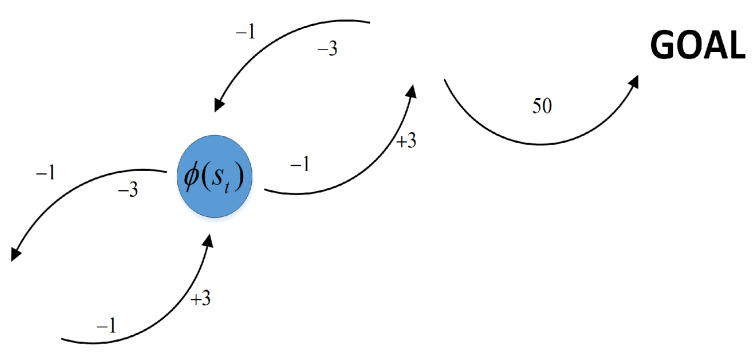
Potential value associated with each state.

**Figure 9 sensors-21-05643-f009:**
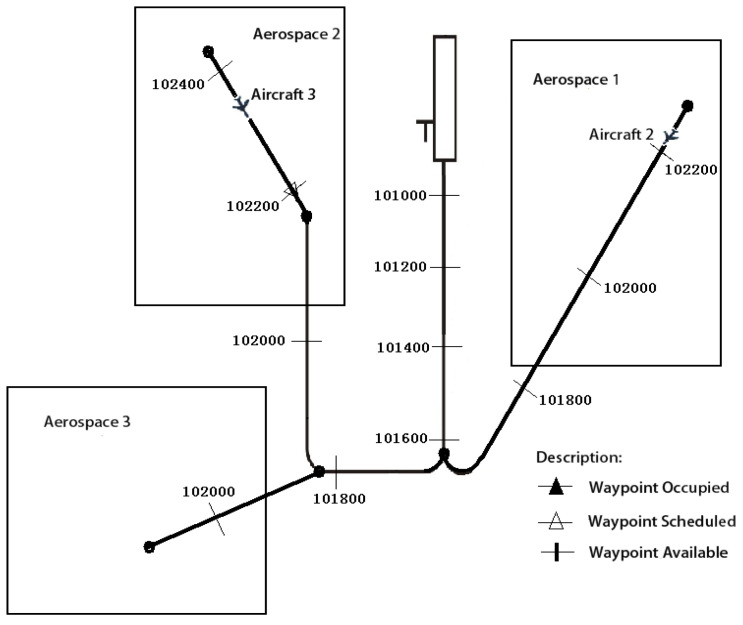
Team aircraft guidance in different aerospaces.

**Figure 10 sensors-21-05643-f010:**
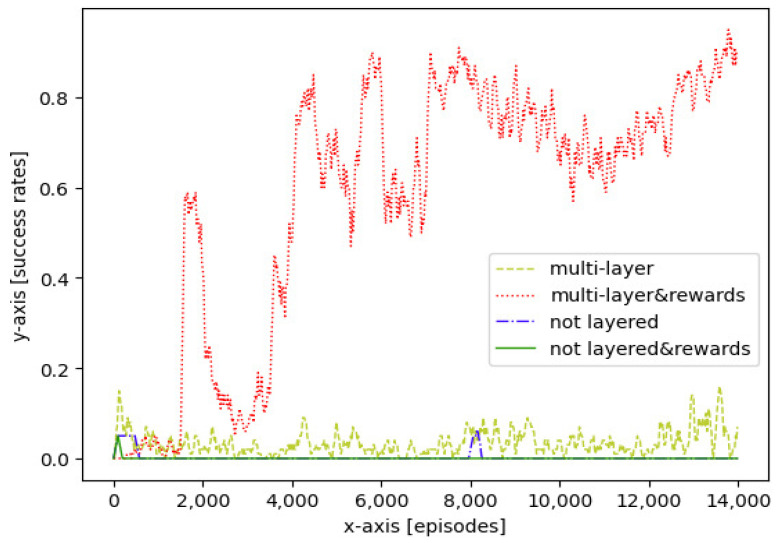
Training success rates, without considering arrival time: multi layer approach without reward shaping; multi layer approach with reward shaping; not layered approach without reward shaping and not layered approach with reward shaping.

**Figure 11 sensors-21-05643-f011:**
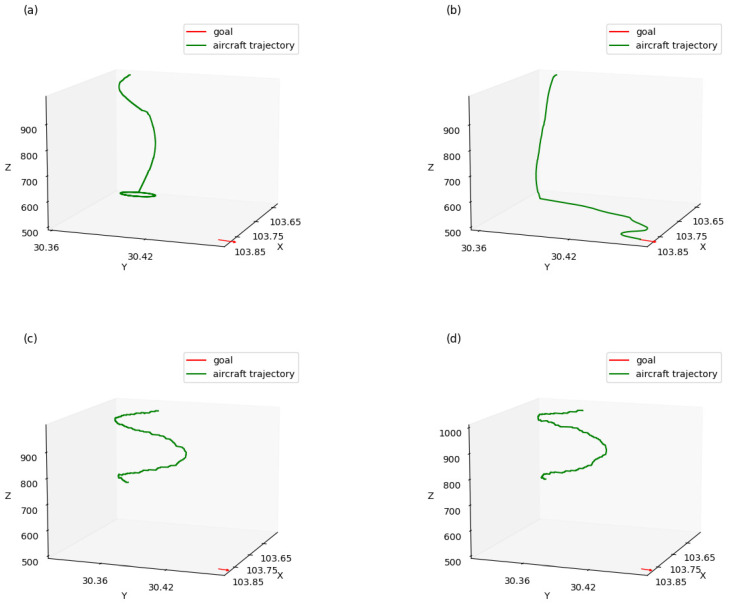
Trajectories of aircraft without considering arrival time: (**a**) multi-layer approach without reward shaping; (**b**) multi-layer approach with reward shaping; (**c**) not layered approach without reward shaping; and (**d**) not layered approach with reward shaping.

**Figure 12 sensors-21-05643-f012:**
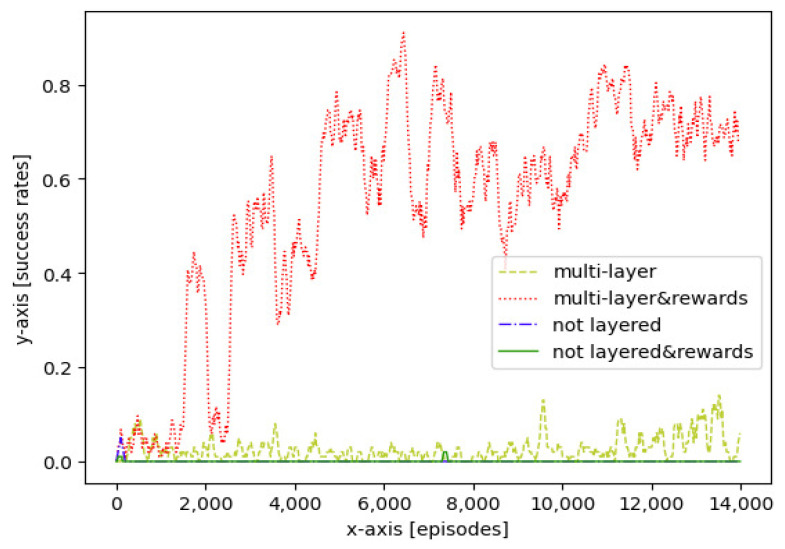
Training success rates, considering arrival time: multi layer approach without reward shaping; multi layer approach with reward shaping; not layered approach without reward shaping and not layered approach with reward shaping.

**Figure 13 sensors-21-05643-f013:**
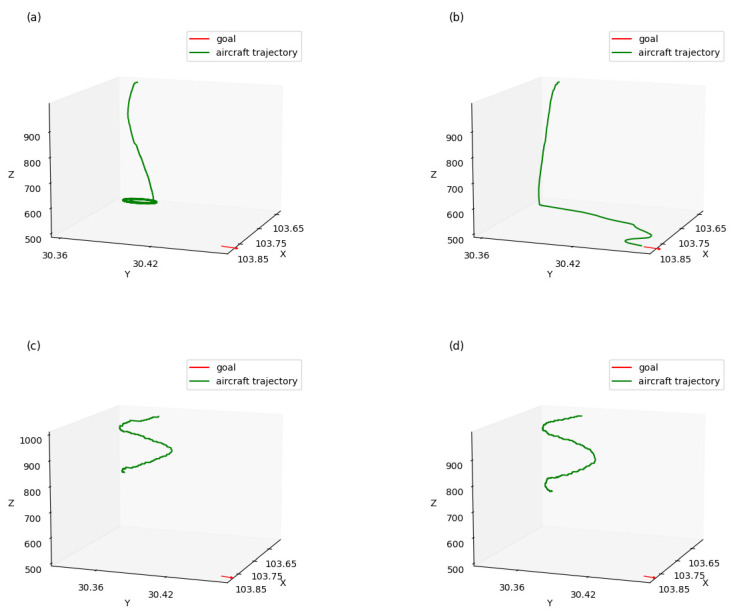
Trajectories of aircraft considering arrival time: (**a**) multi-layer approach without reward shaping; (**b**) multi-layer approach with reward shaping; (**c**) not layered approach without reward shaping; and (**d**) not layered approach with reward shaping.

**Figure 14 sensors-21-05643-f014:**
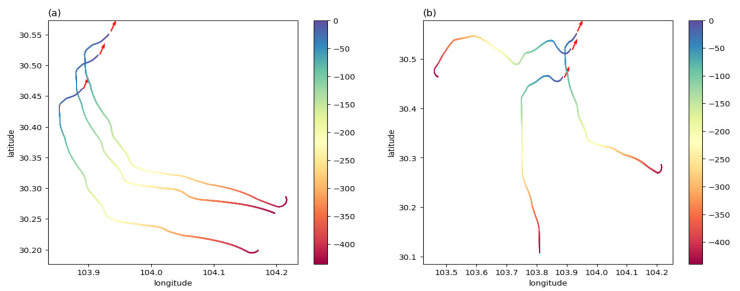
Color maps of aircraft trajectories: (**a**) three aircraft starting in the same aerospace, flying to the aimed for 4D waypoints, respectively, and (**b**) three aircraft starting in different aerospaces, flying to the aimed for 4D waypoints, respectively.

**Figure 15 sensors-21-05643-f015:**
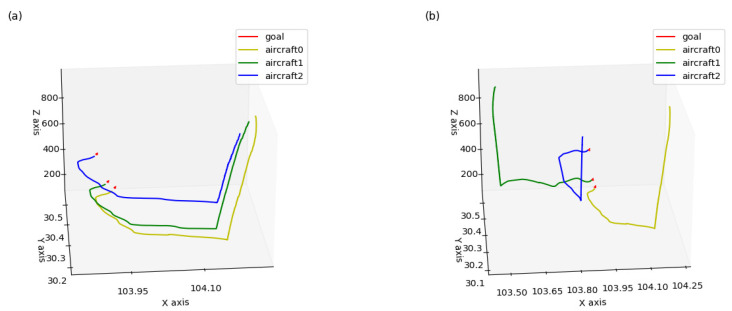
Trajectories of aircraft trained in team aircraft guidance: (**a**) three aircraft starting in the same aerospace, flying to the aimed for 4D waypoints, respectively, and (**b**) three aircraft starting in different aerospaces, flying to the aimed for 4D waypoints, respectively.

**Figure 16 sensors-21-05643-f016:**
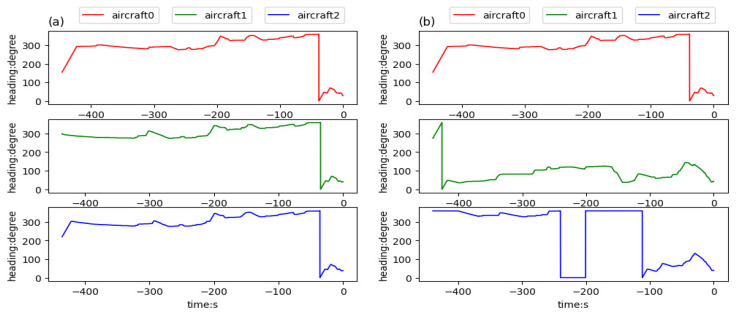
Headings of aircraft trained in team aircraft guidance: (**a**) three aircraft starting in the same aerospace, flying to the aimed for 4D waypoints, respectively, and (**b**) three aircraft starting in different aerospaces, flying to the aimed for 4D waypoints, respectively.

**Figure 17 sensors-21-05643-f017:**
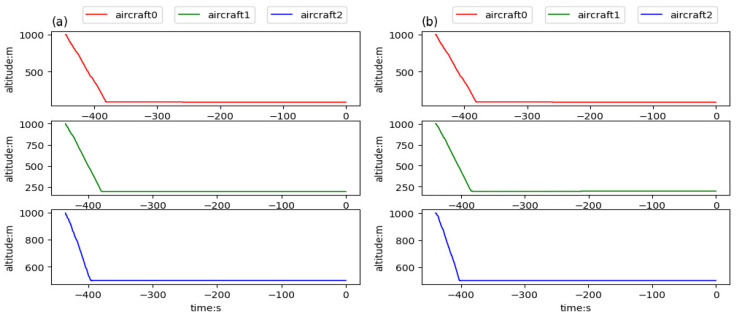
Altitudes of aircraft trained in team aircraft guidance: (**a**) three aircraft starting in the same aerospace, flying to the aimed for 4D waypoints, respectively, and (**b**) three aircraft starting in different aerospaces, flying to the aimed for 4D waypoints, respectively.

**Figure 18 sensors-21-05643-f018:**
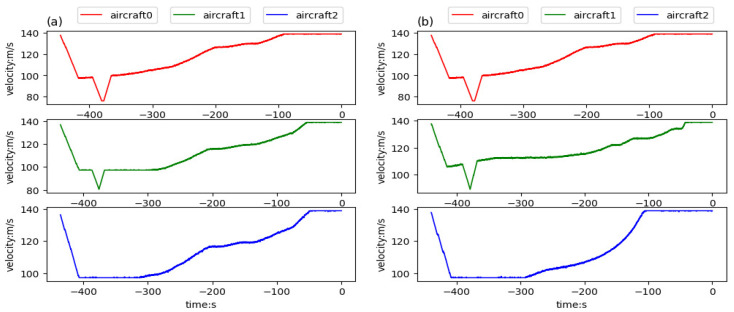
Velocities of aircraft trained in team aircraft guidance: (**a**) three aircraft starting in the same aerospace, flying to the aimed for 4D waypoints, respectively, and (**b**) three aircraft starting in different aerospaces, flying to the aimed for 4D waypoints, respectively.

**Table 1 sensors-21-05643-t001:** Coefficients of reward functions.

Coefficients	Value
$e_{D}$	0.02
$e_{O}$	0.006
$e_{H}$	0.5
$e_{t}$	1
$e_{P}$	0.02

**Table 2 sensors-21-05643-t002:** Simulation environment settings.

Parameters	Value
Airport latitude	$30.5635165^{{}^{\circ}}$
Airport longitude	$103.939946^{{}^{\circ}}$
Airport runway orientation	$22^{{}^{\circ}}$
Aircraft type	F-16
*x* range	[103.4°, 104.5°]
*y* range	[30.1°, 30.6°]
*z* range	[0 m, 1000 m]
Time interval	1 s
Initial horizontal velocity	500 km/h
Maximum horizontal velocity	500 km/h
Minimum horizontal velocity	270 km/h
Maximum climb velocity	3 m/s
Minimum descent velocity	−3 m/s
Maximum turn rate	±6 m/s
Maximum acceleration	±2 m/s
Termination	distance < 2 km and delta heading
	angle < 28° or arrival time < 0

**Table 3 sensors-21-05643-t003:** Hyperparameters of the multi-layer RL model.

Parameters	Value
Replay buffer size	200,000
Discount factor γ	0.98
Learning rate α	$5\times 10^{-4}$
Mini batch size	1000

**Table 4 sensors-21-05643-t004:** Simulation results of algorithm without considering arrival time.

Algorithm	Maximum Success Rate (%)	Average Computational Time (ms)
Multi-layer approach	17	3.4
without reward shaping		
Multi-layer approach	95	3.5
with reward shaping		
Not layered approach	6	2.9
without reward shaping		
Not layered approach	5	2.7
with reward shaping		

**Table 5 sensors-21-05643-t005:** Simulation results of algorithm considering arrival time.

Algorithm	Maximum Success Rate (%)	Average Computational Time (ms)
Multi-layer approach	15	5.5
without reward shaping		
Multi-layer approach	92	5.8
with reward shaping		
Not layered approach	5	3.6
without reward shaping		
Not layered approach	2	3.4
with reward shaping		

**Table 6 sensors-21-05643-t006:** Average distance and delta heading angle to goal.

Born Place	Average Distance (km)	Average Delta Heading Angle (°)
Aircraft in the same aerospace	0.360	13.7
Aircraft in different aerospaces	0.598	14.7

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
