# Peer review of "A Multi-Dimensional Goal Aircraft Guidance Approach Based on Reinforcement Learning with a Reward Shaping Algorithm"

_sensors, 2021, doi:10.3390/s21165643_

Round 1
Reviewer 1 Report
please check the comments file being attached
Reviewer 2 Report
This paper tries to improve the convergence efficiency and trajectory performance by using multi-layer RL with a shaped reward function. The authors present a scheme that uses arrival time along with latitude, longitude, altitude, and arrival time. Overall, the paper is well organized and makes significant progress on guidance simulation.
Comments to authors:
Notations in the formulation are critical for readers; however, some notations were used without any description (for example, P, d, alpha, beta, gamma, V, etc., in the Background section). Some of the notations are redefined over and over in different sections, which confuses the reader. The physical meaning of the loss function described in section 2 is not explained.
In Eq. 8, could you confirm that velocity is used to measure the time to travel pi to p0, and why not speed?
In Eq. 9 and 10, two different notations, Si and si, are used to denote the length of the flight segment Li.
Ji in Eq. 11 is not described. Si or si is defined but was not utilized in other equations.
The authors have compared multi-layer RL without reward shaping against the one with reward shaping; it would have been much better if the authors included the comparisons with other existing models to show the superiority of the presented model.
There are bits of flaws and inconsistencies in the paper that tarnishes the importance of the paper.
Round 2
Reviewer 1 Report
The review process is very satisfactory, beyond expectation.
Authors provide very detailed answers, covering all enumerated issues.
On the technical side, I have no further comments, the notation has been much improved and clarified, extension to the spatial formulation has been provided, the framework parallelism is clearer now in the revised paper.
Please make sure all comments are well integrated into the manuscript.
Please make a language check over again, to maximize readability and user experience.
Ensure also that the references are in the appropriate format.
Author Response
Dear referee,
We thank the referees for the careful reading of our paper. We have carefully considered the comments and revised the manuscript accordingly. Please find below our responses to the reviewers’ comments. All the revisions have been addressed in the Reply and highlighted in the manuscript with orange background. In this revision, we mainly did the language check and references format check. We hope the revised manuscript can be considered acceptable.
Reply to the comments of Reviewer 1
Point 1: Please make sure all comments are well integrated into the manuscript.
Response 1: Thank you for your comment. All comments have been well integrated into the manuscript and have been checked carefully.
Point 2: Please make a language check over again, to maximize readability and user experience.
Response 2: Thank you for your valuable suggestion. We have carefully made the language check over again, and all the revisions are highlighted with orange background in the manuscript.
Point 3: Ensure also that the references are in the appropriate format.
Response 3: We thank you for reminding us this important point. We have modified the format the references as request, by using the format of “BibTex”.
